# Assessing CO₂ Emissions of Online Food Delivery in Japan

**Xujie Hu** [1], **Chen Liu** [2,*] , **Qiannan Zhuo** [1,*] and **Dami Moon** [3]

[1] Graduate School of Media and Governance, Keio University, 5322 Endo, Fujisawa 252-0882, Japan; huxujie@keio.jp

[2] Sustainable Consumption and Production Area, Institute for Global Environmental Strategies, 2108-11 Kamiyamaguchi, Hayama 240-0115, Japan

[3] Department of Urban Engineering, School of Engineering, The University of Tokyo, 7-3-1, Hongo, Bunkyo, Tokyo 113-8656, Japan

[*] Correspondence: c-liu@iges.or.jp (C.L.); s12381ss@keio.jp (Q.Z.)

**Abstract:** The COVID-19 pandemic and the subsequent lockdown of cities have led to the rapid growth of online food delivery (OFD). Moreover, there are concerns that OFD platforms may impose offers on users in order to continue to increase their market share, leading to numerous environmental issues such as overconsumption and a significant increase in plastic packaging waste. Most studies have focused on the environmental impacts associated with food packaging and have been mostly limited to China. However, less research has been carried out on the overall CO₂ emissions of an OFD order including food. In this study, the CO₂ emissions of an OFD order were assessed by considering the production, distribution, consumption and disposal of the ingredients, based on lifecycle thinking and existing secondary data, for three representative food groups (Western food, Japanese food and Chinese food) in Japan. This study found that the food production of an OFD order accounts for more than 70% of the CO₂ emissions of the entire process, especially food ingredient production. Policy support and initiatives such as OFD platforms being able to serve different quantities of food based on actual consumer demand to avoid food waste, as well as changes in delivery methods, would help reduce the CO₂ emissions of OFD.

**Keywords:** online food delivery; life cycle thinking; CO₂ emissions; Japan

## 1. Introduction

### 1.1. Background

Explosive growth in the online food delivery (OFD) industry has been observed in recent years. Internet platforms allow customers to buy food from partnering restaurants through smartphone applications and have it delivered to their households [1]. The rapid uptake of OFD in the last decade has been attributed to factors such as the growth of the internet and e-commerce, hedonic motivation, increasing household discretionary income and convenience and time-saving in increasingly busier lives [2,3].

The COVID-19 pandemic has accelerated OFD development in Japan over the past two years. According to Measurable AI's e-receipts data, Japan's OFD market share comprises two major companies from 2020 to 2021: Uber Eats (market share of over 60%) and Demae-Can (market share of about 30%) [4]. The OFD industry increased by 25% from 2016 to 2020, and it is set to increase by a further 17% from 2021 to 2025 [5].

There have been many studies on OFD in recent years, mainly focusing on obesity [6–8], cloud kitchens [9,10], food safety issues [11], consumption habits [12,13], social relationships [14] and traffic accidents [15–17]. However, the impact of OFD comes from three main areas: economic, social and environmental. In social terms, OFD has greatly increased food accessibility, but it has also greatly increased the availability of unhealthy food [18]. This is because OFD platforms have a large selection of dishes with low nutritional value. OFD provides work opportunities for more people and greatly enhances

work flexibility. However, in some countries, couriers have few employment rights and are subject to a dangerous working environment [19]. On an economic level, OFD's market revenue was boosted by 27% to USD 136.4 billion in 2020 [20]. Furthermore, the development of OFD brings benefits to many industries, such as the food packaging industry, manufacturing (e-bikes) and sales. However, there have also been some negative effects on the traditional restaurant industry, with many restaurants having to adapt their business models to stay afloat [21]. Moreover, OFD platforms, through aggressive marketing and promotional strategies, may lead to over-consumption by consumers [18]. This behavior will greatly increase the burden on waste disposal. As for the environmental impact, the most important point is the generation of large amounts of plastic packaging and how to deal with it [21]. For example, the amount of waste from OFD packaging in China increased by 650% in 2017 compared to 2015 levels [22]. In 2020, influenced by COVID-19, consumers preferred single-use plastic packaging, which in turn led to a rapid growth in plastic packaging waste [21]. Another important environmental issue is $CO_2$ emissions associated with OFD. The main sources of environmental impact (solid waste pollution, water pollution, recourse consumption, and air pollution) are production and waste disposal along the whole industry chain of OFD packaging [23].

Since the OFD industry is relatively young, research and discourse on OFD and its environmental effects is limited. In addition, many studies have focused on China, as OFD was growing rapidly in China before the COVID-19 pandemic. Meanwhile, due to the impact of the pandemic, the OFD industry has grown rapidly in Japan. Meals from restaurants/fast food chains will facilitate the next increase in OFD over the next five years [4,5]. This makes OFD research in Japan a new academic hotspot.

### 1.2. Unresolved Issues

Current research on the $CO_2$ emissions of OFD has focused on the $CO_2$ emissions of OFD-related packaging from production to disposal, and researchers have found that the production of raw packaging materials accounts for at least 50% of the entire process. However, due to the complexity and limitations of the food data, most studies do not consider the contribution of food ingredients in the $CO_2$ emissions of OFD. Moreover, distribution, an important part of OFD, is not addressed in many studies. Based on secondary data, this study analyzes the $CO_2$ emissions of OFD from three different types of food (Western food, Japanese food and Chinese food) in Japan, taking food production in particular into account.

### 1.3. Research Objectives

Existing studies have only focused on $CO_2$ emissions from OFD-related packaging and have not considered the $CO_2$ emissions of food production and distribution. This study considered Western food, Japanese food and Chinese food as three representative OFD food groups and developed a framework to estimate the overall greenhouse gas emissions of an OFD order through an entire process by considering the production, distribution, consumption and disposal of the food ingredients and related plastics, based on lifecycle thinking and existing secondary data in Japan. Through scenario analysis, the framework not only compared the impact of three different food groups on the $CO_2$ emissions of OFD, but also calculated the contribution of the different distribution method and disposal method to the $CO_2$ emissions of OFD. Furthermore, policy recommendations to prevent and reduce $CO_2$ emissions are provided based on the scenario analysis.

## 2. Literature Review

The relevant literature and the main findings on $CO_2$ emissions from OFD and on consumer behavior have been reviewed as follows.

### 2.1. $CO_2$ Emissions from OFD

The increase in OFD will affect all stages of the food supply chain and have more serious environmental issues. Significant $CO_2$ emissions have occurred from food packaging production, food delivery and waste generation [21,24]. Two-thirds of packaging demand comes from the food industry, such as food containers, cutlery, napkins and plastic bags, which will lead to resource depletion and large $CO_2$ emissions due to the high demand for energy and raw materials [21]. These materials are frequently single-use, necessitating huge amounts of energy and raw materials to manufacture, transport and dispose of them. Packaging production and disposal can also lead to the release of a wide range of pollutants into the environment [25]. All of this points to the OFDs catalyzing the increasing consumption of packaging-derived plastics, which already account for 46% of worldwide plastic waste.

Research on OFD has mainly focused on $CO_2$ emissions related to different OFD product packaging processes, from production to waste treatment [1,26], and $CO_2$ emissions related to OFD waste disposal [27]. For example, Arunan et al. [1] quantified the packaging-related greenhouse gas emissions associated with OFD orders in Australia. Packaging raw materials contribute at least 50%. As the OFD industry continues to grow, $CO_2$ emissions associated with OFD packaging are expected to increase by 132% by 2024. Liu et al. [28] found that OFD waste is a small proportion of municipal solid waste; however, food packaging waste accounts for 15.7% of the total. Plastic bags were the most used packaging at 35%, and plastic boxes accounted for 27%. However, the environmental impact of paper boxes cannot be ignored. Despite their lower usage, the environmental impact potential of NOx generated during the production of paper boxes far exceeds that of $CO_2$. Camps-Posino et al. [29] assessed the impact of OFD's packaging and its waste disposal on climate change based on LCA and explored the advantages of increasing recycling rates, the amount of packaging recycled and the use of reusable packaging. This study found that takeaway packaging in China emits approximately 13 million tons of $CO_2$. If the recycling rate is increased to 35%, packaging emissions would be reduced by 16%. If packaging made from recyclable materials is increased by 50%, emissions would be reduced by 60%. In addition, if reusable packaging is used, emissions would be reduced by 63%. Zhang et al. [27] combined the direct weighing method with a questionnaire to analyze the scale, pattern and impact of OFD waste. This research found that the total amount of OFD waste in 2019 was 177.6 kilotons, $CO_2$ emissions were 168.3 kilotons and packaging waste accounted for 32%. Significant differences in OFD waste were found between different consumer groups, with white-collar workers contributing the highest amount at 58%. This could be reduced by 25% if waste-to-energy technologies were implemented as planned, and by 55% if the avoidable portion of OFD waste was avoided. Some researchers found that recycling these packages could significantly ease energy consumption for production, thereby reducing $CO_2$ emissions [24,30]. In Australia, some food companies offer reusable food containers (stainless-steel containers), greatly enhancing the sustainability of food packaging [1].

### 2.2. Consumer Behavior Impacts

The impact of consumer behavior on OFD comes from two main components, purchasing behavior and the selection of the disposal of OFD waste.

For consumers, there is no doubt that selecting OFD can save them a great deal of time. In China, each OFD order can save consumers at least 48 min [21]. Studies have also shown that OFD produces less food waste than eating at home, because home cooking sometimes has led to an increase in food waste by producing more food than household demand [31]. The relatively small amount of food, due to the limitations of the container, makes it easier for consumers to finish their meal. Furthermore, $CO_2$ emissions from the production and preparation of different foods are different. For example, in Japan, a study found that Western food has higher $CO_2$ emissions than Chinese food or Japanese food in the household [32]. So, from an environmental point of view, the kind of food that

consumers choose determines the carbon footprint of OFD at the production stage, and selecting OFD can reduce a certain amount of food waste.

Consumer attitudes are an important factor in evaluating the environmental impact of the food supply chain, particularly in waste disposal and recycling [33]. Some studies have found that changes in consumer behavior at the waste disposal stage can significantly reduce $CO_2$ emissions [34]. Despite consumers' environmental awareness, most people are unaware of how food waste is generated, as well as recooking leftovers and reusing meal containers in the future [12]. For example, in Changchun, China, more than half of university students do not separate leftover food from its container and throw it away. The main reason for this is a lack of knowledge about waste separation [35]. While there are many factors that influence consumer sorting behavior, such as age, gender, education level, income level and the uneven distribution of waste facilities [36], for those consumers who are not interested in recycling, the inconvenience of disposal and recycling is the main reason [1].

## 3. Materials and Methods

### 3.1. Scope and Unit Definition

The scope of this research is to assess the environmental impact of online delivery. As shown in Figure 1, this study divides the environmental impact of OFD into four components: production, distribution, consumption and end of life/recycling. The environmental impact indicators for assessing OFD mainly consider $CO_2$ emissions and waste volumes. For this research, the functional unit is defined as a single packaged online food order for one person.

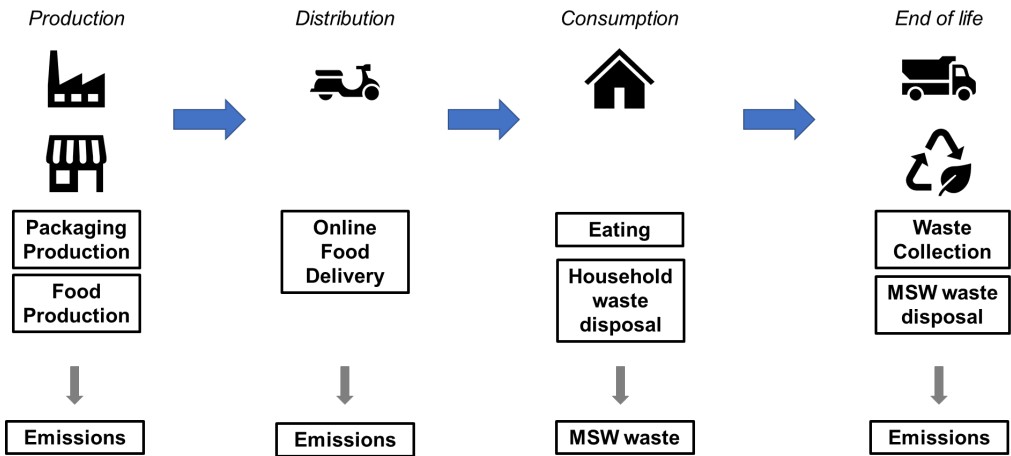

**Figure 1.** System boundaries.

### 3.2. Calculation Framework

Figure 2 shows the calculation framework for OFD consisting of the production stage, distribution stage, consumption stage and waste disposal stage. The red rectangle represents the target, the green rectangle represents the calculation process and the dotted line represents no data. In production, this stage not only considers food ingredient production and OFD container production, but also considers cooking in restaurants. In distribution, this stage contains a wide range of transport options, such as cycling, motorized bicycles, motorbikes and vehicles. Waste disposal considers not only waste collection and transportation but also landfill and incineration.

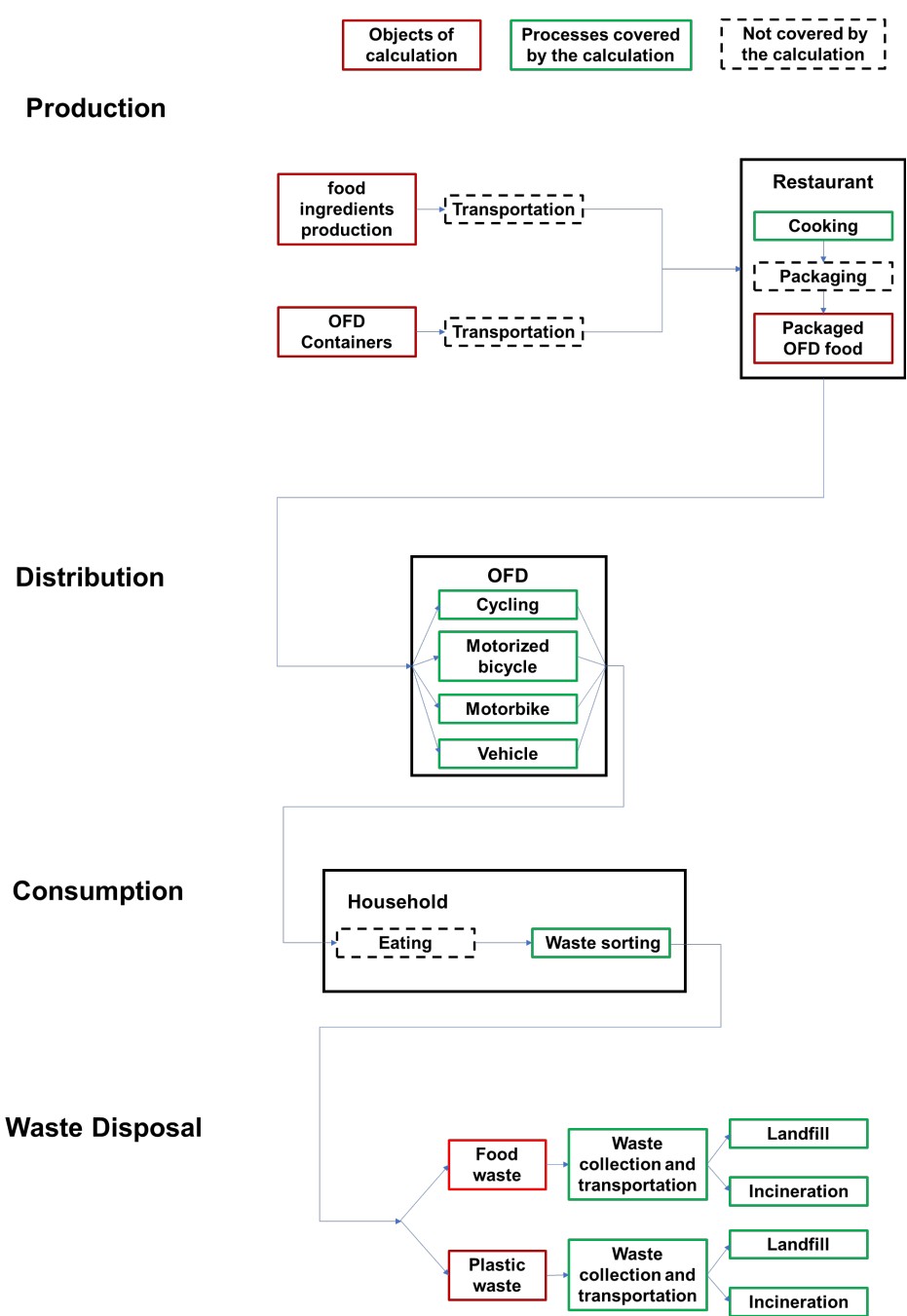

**Figure 2.** Calculation framework.

The framework analyzes the environmental impacts of different OFDs and the impacts of different consumer behaviors on OFD. Firstly, the contribution of $CO_2$ emissions at different LCA stages of different OFDs is analyzed. Secondly, the environmental impacts associated with consumer decisions regarding OFD purchase, consumption and disposal are analyzed.

*3.3. Databases*

3.3.1. Food Production

In this research, food production includes food ingredient production and cooking in the restaurant. Table 1 shows the $CO_2$ emissions of food ingredient production and the household cooking of three types of food, where food ingredient production includes the processes of ingredient production, packaging and transportation [32]. $CO_2$ emissions from

restaurants are 1.2 to 1.8 times higher than household emissions, averaging 1.5 times [37]. Table 2 shows the LCA progress of $CO_2$ emissions of food cooking in restaurants.

**Table 1.** $CO_2$ emissions of food production in households.

| Food Types | Materials | $CO_2$ of Food Ingredient Production (kg-$CO_2$/per Meal) | $CO_2$ of Food Cooking (kg-$CO_2$/per Meal) |
|---|---|---|---|
| Western food | Bread | 0.04303 | 0.00770 |
| | Hamburg steak | 1.12801 | 0.04911 |
| | Potato salad | 0.10577 | 0.03484 |
| | Vegetable soup | 0.11105 | 0.01177 |
| Japanese food | Rice | 0.04734 | 0.02012 |
| | Tempura | 0.38877 | 0.00011 |
| | Pickles | 0.10921 | 0.00000 |
| | Miso soup | 0.05581 | 0.01177 |
| Chinese food | Rice | 0.04734 | 0.02012 |
| | Dumplings | 0.17401 | 0.01705 |
| | Fried vegetables | 0.24929 | 0.01783 |
| | Chinese soup | 0.15694 | 0.01177 |

**Table 2.** Total $CO_2$ emissions of food production in households and restaurants.

| Food Types | $CO_2$ Emissions in Household (kg-$CO_2$/per Meal) | $CO_2$ Emissions in Restaurant (kg-$CO_2$/per Meal) |
|---|---|---|
| Western food | 1.491 | 2.236 |
| Japanese food | 0.633 | 1.010 |
| Chinese food | 0.694 | 1.041 |

3.3.2. Food Weight

Based on previous studies [38,39], the different components were weighed. Compared to Western and Japanese food, Chinese food is the heaviest. This is because the raw materials for Chinese soups come mainly from large amounts of chicken and water, whereas Western soups are based on small amounts of vegetables and milk. As shown in Table 3, compared with Western food and Japanese food, the containers for Chinese food are complex.

**Table 3.** Weight of three food types.

| Food Types | Materials | Weight (kg) | Images |
|---|---|---|---|
| Western food | Total | 0.585 |  |
| | Bread | 0.06 | |
| | Hamburg steak | 0.289 | |
| | Potato salad | 0.129 | |
| | Vegetable soup | 0.108 | |
| Japanese food | Total | 0.750 |  |
| | Rice | 0.100 | |
| | Tempura | 0.359 | |
| | Pickles | 0.030 | |
| | Miso soup | 0.261 | |
| Chinese food | Total | 0.885 |  |
| | Rice | 0.100 | |
| | Dumplings | 0.060 | |
| | Fried vegetables | 0.324 | |
| | Chinese soup | 0.401 | |

### 3.3.3. Container Production

We obtained some common materials and weights of OFD containers from a website [40] (Table 4). The selection of container styles is based on the images in Table 3.

**Table 4.** Weight of container.

| Food Types | Materials | Weight (kg) |
|---|---|---|
| Japanese food | PSP | 0.007 |
| | PSP | 0.005 |
| | PSP | 0.005 |
| | PP | 0.005 |
| Western food | PSP | 0.015 |
| | PSP | 0.005 |
| | PP | 0.005 |
| Chinese food | PSP | 0.005 |
| | PP | 0.005 |

(PP: polypropylene; PSP: polystyrene paper).

This study describes the $CO_2$ emissions from containers made of different materials (Table 5) [41]. For environmentally friendly materials (Bio-PE, PLA and PHBH), $CO_2$ emissions data from the manufacturing process are not available and are not considered in this study.

**Table 5.** $CO_2$ emissions of container material.

| Materials | $CO_2$ Intensity in Production Process (kg-$CO_2$/kg) | $CO_2$ Intensity in Manufacturing Process (kg-$CO_2$/kg) | Total $CO_2$ Intensity (kg-$CO_2$/kg) |
|---|---|---|---|
| PSP | 2.695 | 0.56 | 3.255 |
| PP | 1.52 | 1.013 | 2.533 |

(PP: polypropylene; PSP: polystyrene paper).

### 3.3.4. Distribution of OFD

A survey conducted by Tokyo Smart Restaurant LLC in 2022 of 1013 takeaways found that delivery was mainly carried out using bicycles, motorized bicycles, motorbikes and vehicles [42]. Table 6 shows the $CO_2$ emissions from OFD's main distribution [43]. Within the framework of the system we designed (Figure 2), the distribution of OFD is only considered from the restaurant to the home, and the main $CO_2$ emissions from this process are mainly from energy consumption.

**Table 6.** $CO_2$ emissions of distribution.

| Types | $CO_2$ Emissions (kg-$CO_2$/Person·km) |
|---|---|
| Cycling | 0 |
| Motorized bicycle | 0.031 |
| Motorbike | 0.092 |
| Vehicle | 0.190 |

### 3.3.5. Waste Disposal

According to the IDEA database, this study established a waste disposal database (Table 7). It includes landfill services and incineration services without power generation.

**Table 7.** $CO_2$ emissions of waste treatment.

| IDEA Product Code | Product Name | Unit | kg-$CO_2$ |
|---|---|---|---|
| 851611201 | Landfill treatment service, domestic waste, waste plastics | kg | 0.0357 |
| 851611202 | Landfill treatment service, domestic waste, kitchen garbage | kg | 0.0628 |
| 851612000 | Incineration service, domestic waste | kg | 0.9751 |
| 851612201 | Incineration service, domestic waste, waste plastics | kg | 2.9745 |

*3.4. Impact Assessment*

3.4.1. Quantifying the $CO_2$ Emissions of Packaging Production

Combining Tables 5 and 6, the environmental impacts of food packaging were estimated. The $CO_2$ emissions of OFD packaging in the production progress can be calculated using Equation (1):

$$C_p = O_p \times W_{po} \tag{1}$$

where $C_p$ is the $CO_2$ emissions from production food packaging, $O_p$ is the carbon intensities, and $W_{po}$ is the weight of the food package per order.

3.4.2. Quantifying the $CO_2$ Emissions of Distribution

If we want to quantify the delivery-related $CO_2$ emissions from restaurant to household, we need to consider two important factors: the type of delivery and the distance from the restaurant to the household. According to the Demae-Can website, we found that the average delivery distance is 1.7 km [44]. Therefore, according to Table 6, we could calculate the $CO_2$ emissions of different types of OFDs.

3.4.3. Quantifying the $CO_2$ Emissions of Waste Treatment

The annual $CO_2$ emissions of the waste treatment of OFD packaging waste can be calculated using Equations (2) and (3) [45]:

$$C_{ct} = W_{po} \times c_{ct} \tag{2}$$

$$C_{wt} = W_{po} \times c_{wt} \tag{3}$$

where $C_{ct}$ is the total $CO_2$ emissions of the waste collection and transportation (kg $CO_2$); $C_{wt}$ is the total $CO_2$ emissions of the OFD waste treatment (kg $CO_2$); $c_{ct}$ is the GHG emission intensity of the waste collection and transportation (kg $CO_2$/kg); $c_{wt}$ is the GHG emission intensity of the waste treatment (kg $CO_2$/kg).

*3.5. Scenario Analysis*

Consumers' dietary choices and waste disposal will largely influence the environmental impact of OFD. As a result, scenario analysis was used to test it in the framework's last section under diverse consumer behavior patterns. The frameworks of scenarios representing various customer behavior patterns are shown in Figure 3. Food and packaging selection, distribution, distance, food waste, packaging disposal, trash transportation and plastic treatment are the seven pattern steps investigated.

Combinations of patterns from the nine stages are used to illustrate the scenarios. A eight-digit number combination is used to identify each food product situation. Each digit in the scenario's nomenclature corresponds to a pattern in eight of the nine stages, excluding packaging disposal patterns. For example, one scenario is designated as "12321142" when food selection is Western food (pattern 10000000), packaging is PP (pattern 1000000), distribution is by motorbike (pattern 300000), distance is 1–2 km (pattern 20000), food waste is 0 (pattern 1000), the weight of the soup is ignored, waste transportation is pattern 100, food disposal is not needed (pattern 30) and plastic disposal is via incineration (pattern 2). Waste treatment is determined by the consumer when they dispose of food waste and

packaging. Due to the lack of recycling data, this model calculation only considers landfill and incineration.

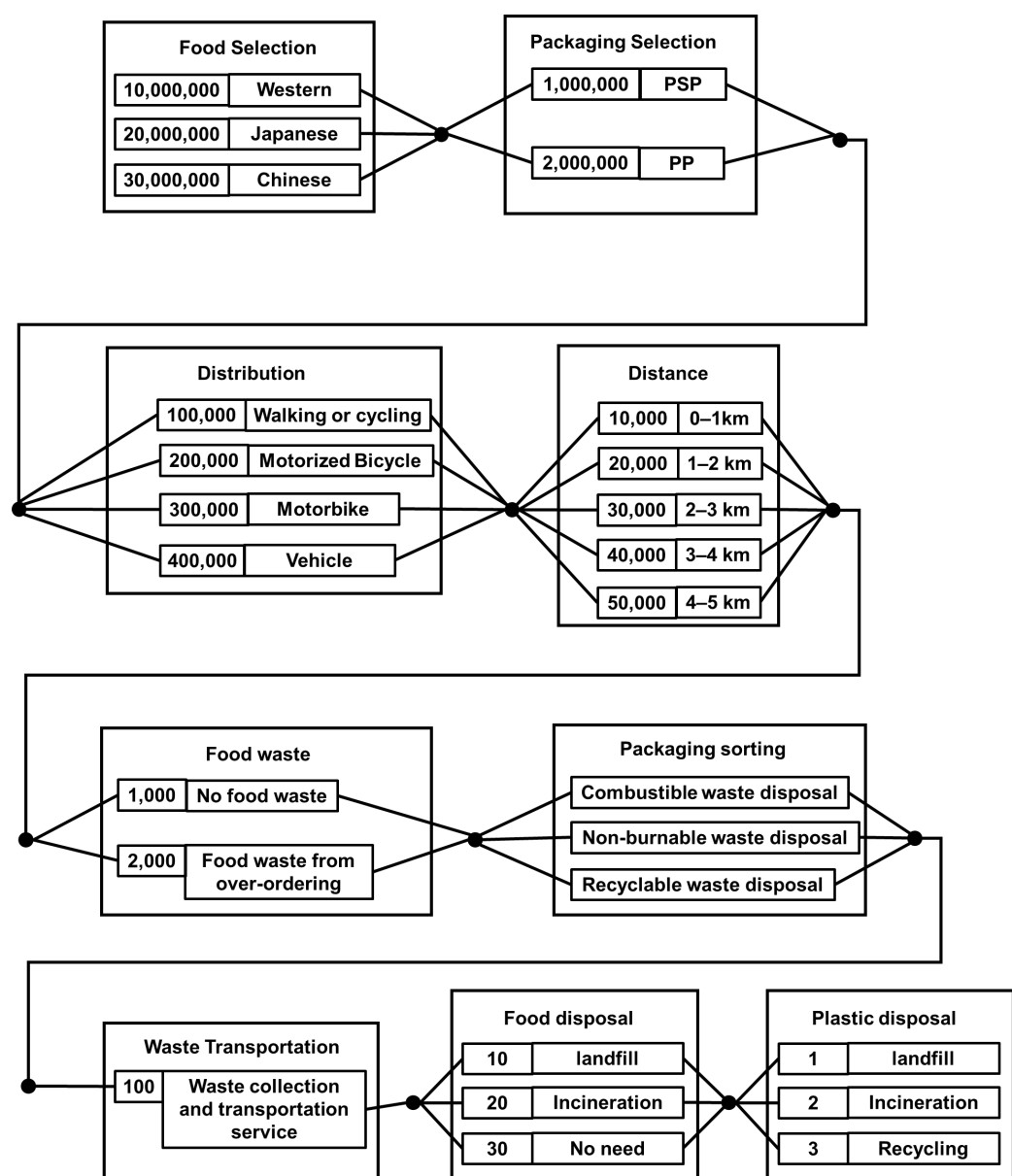

**Figure 3.** Scenario analysis from food selection to plastic disposal.

## 4. Results

### 4.1. CO$_2$ Emissions from OFD

Figure 4 shows the CO$_2$ emissions from three typical OFD foods. The selection of food and plastic containers is determined by the actual situation. According to the questionnaire on the iideli website [46], during the pandemic, 69.7% of takeaways chose motorbikes, so motorbike was chosen as the distribution. According to the Demae-Can website, the average delivery distance is 1.7 km [44]. Because the functional unit was set up as a one-person takeaway, we assumed that the consumer ate all the food and did not produce food waste. According to the Fundamentals of Plastics Recycling report [47], 79.5% of municipal waste (included plastic waste) is incinerated, so the form of plastic waste disposal was chosen as incineration.

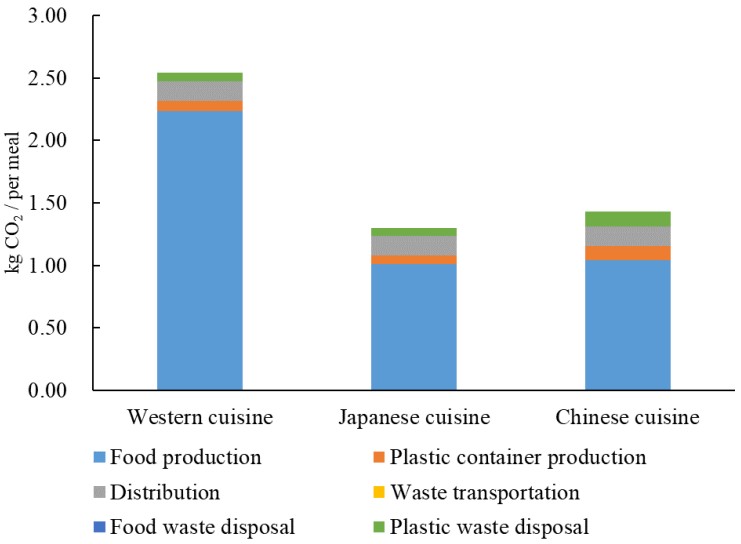

Scenario description

| Scenario label | 1(12)321132 | 2(12)321132 | 3(12)321132 |
|---|---|---|---|
| Food selection | Western cuisine | Japanese cuisine | Chinese cuisine |
| Container selection | PSP+PP | | |
| Distribution | Motorbike | | |
| Distance | 1.7 km | | |
| Food waste | 0% waste | | |
| Container disposal | Combustible | | |
| Waste transportation | Waste collection and transportation | | |
| Food waste disposal | No need | | |
| Plastic waste disposal | Incineration | | |

**Figure 4.** $CO_2$ emissions of OFD.

We can find that Western cuisine has the highest $CO_2$ emissions, followed by Chinese and Japanese cuisines. Food production accounts for 87.9%, 77.7% and 72.7% of the total CO2 emissions, respectively. On the other hand, plastic container production only accounts for 3.1%, 5.2% and 8.1% of the total emissions, and all $CO_2$ emissions related to plastic, including plastic manufacturing, the production of plastic containers, plastic waste collection and transportation and disposal of waste, account for 6.0%, 10.3% and 16.4%, respectively. Therefore, it has been found that food production has a significant impact on the $CO_2$ emissions of OFD. On the other hand, the option of OFD container materials does not have a significant environmental impact on OFD since the total weight per meal is not high.

*4.2. The $CO_2$ Emissions of Distributions*

Figure 5 illustrates the best, average and worst scenarios for $CO_2$ emissions from OFD in Japanese food. According to the Japan Food Delivery Company report [48], the most popular OFD food is Japanese food, accounting for 88%, so Japanese food was selected as the food. The material of food packaging, depending on the actual situation, is HIPS. Due to the hilly terrain in Japan, OFD cannot rely entirely on bicycle delivery. Therefore, the appropriate distribution method was chosen for different areas and distances. We roughly assumed 0–2 km distances were traveled by cycling, 2–4 km distances were covered with motorized bicycles or motorbikes, and distances of over 4 km were covered with motorbikes or other vehicles for the calculation. Therefore, the best scenario was to choose cycling for a distance of 0–1 km. The average conditions would be a motorbike and a distance of 1.7 km. According to the Demae-Can APP, the maximum distance found for OFD was about 5 km,

so the worst scenario was a vehicle and a distance of 5 km [44]. The selection of food waste, packaging disposal, waste transportation, food treatment and plastic treatment are the same results as 3.1.

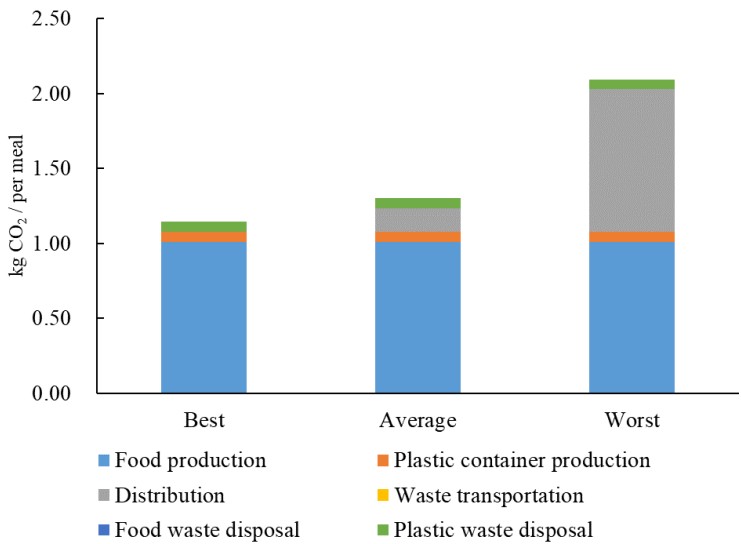

Scenario description

| Scenario label | 1(12)111142 | 1(12)321142 | 1(12)451142 |
|---|---|---|---|
| Food selection | Japanese food | | |
| Container selection | PSP+PP | | |
| Distribution | Cycling | Motorbike | Vehicle |
| Distance | 1 km | 1.7 km | 5 km |
| Food waste | 0% waste | | |
| Container disposal | Combustible | | |
| Waste transportation | Waste collection and transportation | | |
| Food waste disposal | No need | | |
| Plastic waste disposal | Incineration | | |

**Figure 5.** $CO_2$ emissions of different distributions for Japanese food.

In the different conditions of each scenario, the distribution stage contributes from 0% to 45.4% of the entire $CO_2$ emissions. Cycling delivery does not generate any $CO_2$ emissions. However, when the customer lives far from the restaurant and gasoline vehicles need to be used for delivery, $CO_2$ emissions for the distribution stage suddenly make up a large share. Although the environmental impact of OFD distribution can be small or large, the share of vehicle distribution in Japan is only 3.6%, with 94.7% of distribution coming from bicycles and motorbikes. Overall, the environmental impact of OFD distribution ranges from 0 to 12.0%.

## 5. Discussion

### 5.1. Over-Ordering

The results of this study are based on the Japanese Food Guide Spinning Top for food intake per person per meal [49]. However, many businesses in OFD platforms set minimum spending amounts, resulting in consumers having to buy 1.5–2 times more food. Therefore, Figure 6 is based on the scenario design of result 4.1 and adds a second salad, tempura and dumplings to the Western, Japanese and Chinese meals, respectively, to meet the minimum spending requirement. However, for health reasons, the extra food (second salad, tempura and dumplings) is not eaten and ends up being disposed of as food waste. According to the latest Japan Waste Disposal Report that was published by the Ministry of

the Environment, Japan [50], 224 out of 1741 municipalities collected food waste separately from burnable waste, which means the remaining 1517 municipalities put food waste into incineration facilities together with burnable waste. Therefore, food waste disposal is carried out via incineration.

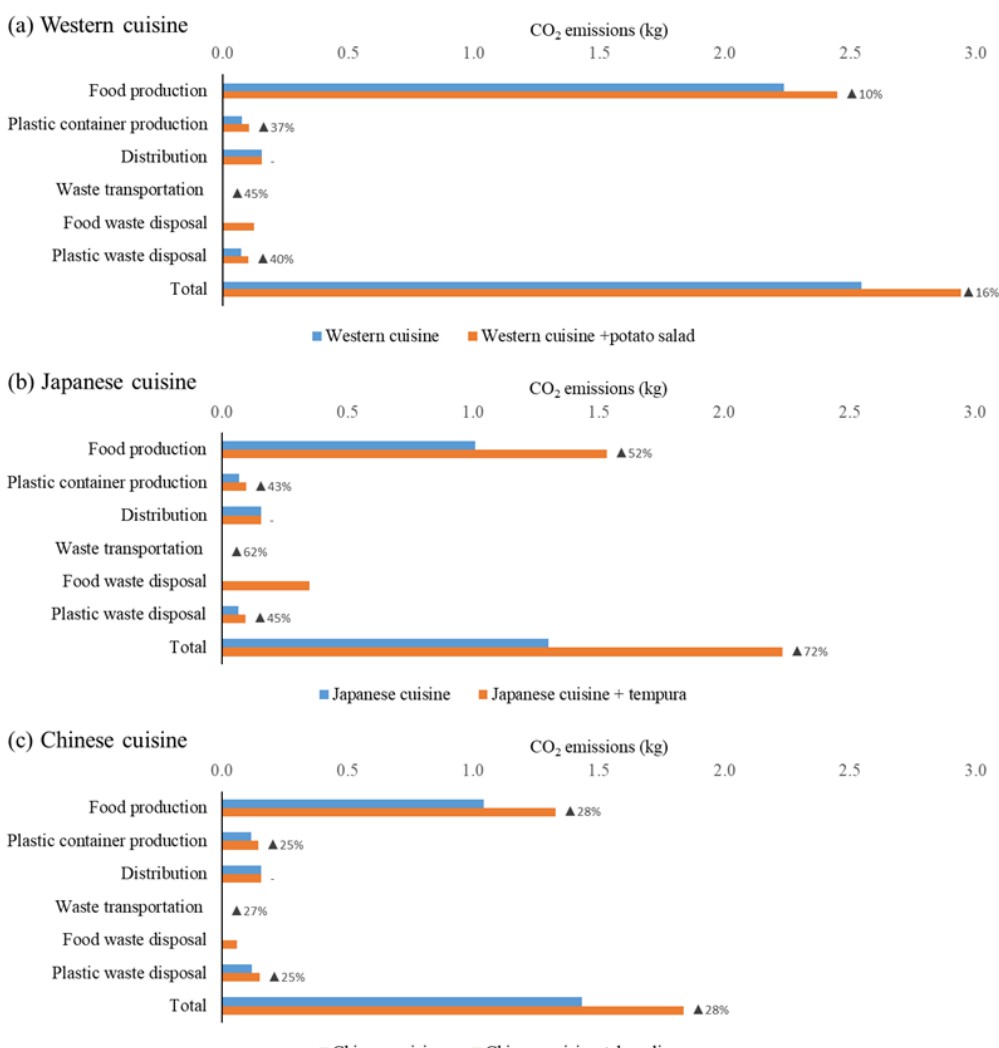

**Figure 6.** Differences in $CO_2$ emissions of OFD by over-ordering.

Figure 6 shows the excess $CO_2$ emissions triggered by the increase in food leftovers due to over-ordering at different stages of OFD. We can find that additional food led to an increase of 16%, 72% and 28% in the total $CO_2$ emissions of OFD for Western cuisine, Japanese cuisine and Chinese cuisine, respectively. Meanwhile, over-ordering food leads to an increase in $CO_2$ emissions at different stages of OFD, especially food waste disposal. Taking Japanese cuisine as an example, the second tempura led to an increase in $CO_2$ emissions of 52%, 43%, 62% and 45% in food production, plastic production, waste transportation and plastic waste disposal. For food waste disposal, $CO_2$ emissions increased from 0 to 15.7% of total emissions. Therefore, food leftovers due to over-ordering have a significant impact on the $CO_2$ emissions of OFD.

Food wastage as a result of OFD is frequently associated with the "minimum price" set by restaurants on the OFD platform, which leads to users buying more food than they need to satisfy the "minimum price" of the free delivery service [12]. This pattern of minimum prices leads to the creation of leftovers for consumers. The inability to preserve leftovers in a timely manner or the reluctance of consumers to preserve them leads, on the one hand, to an increase in food waste, visible to the consumer, and on the other hand, to an increase in

$CO_2$ emissions due to over-ordering, especially at the waste collection and disposal stage, which is a part of the impact that is invisible to the consumer. If OFD platforms could offer different amounts of food options to meet the different needs of different consumers, it would not only reduce food waste but also significantly reduce $CO_2$ emissions.

### 5.2. Shifting the Distribution

Although the environmental impact of distribution for each order is not high, the question remains as to whether the environmental impact of distribution for OFD is also not significant at urban or national scales due to the increasing demand. According to public data, we discovered that the total number of orders in 2019 from Demae-Can was 3.25 million orders, and the current shares of delivery methods were as follows: bicycle (25%), motorbike (69.7%), vehicle (3.6%) and other (3.6%) [44].

Therefore, we have simply estimated the environmental impact of Demae-Can's distribution at the national scale in 2019 (Figure 7). The original pattern is the total $CO_2$ emissions of Demae-Can's distribution in 2019 with a delivery distance of 1.7 km [44]. The S1 pattern replaces 50% of motorcycle with a motorized bicycle while keeping everything else unchanged. The S2 pattern replaces all motorcycles with a motorized bicycle and replaces all cars with motorcycles.

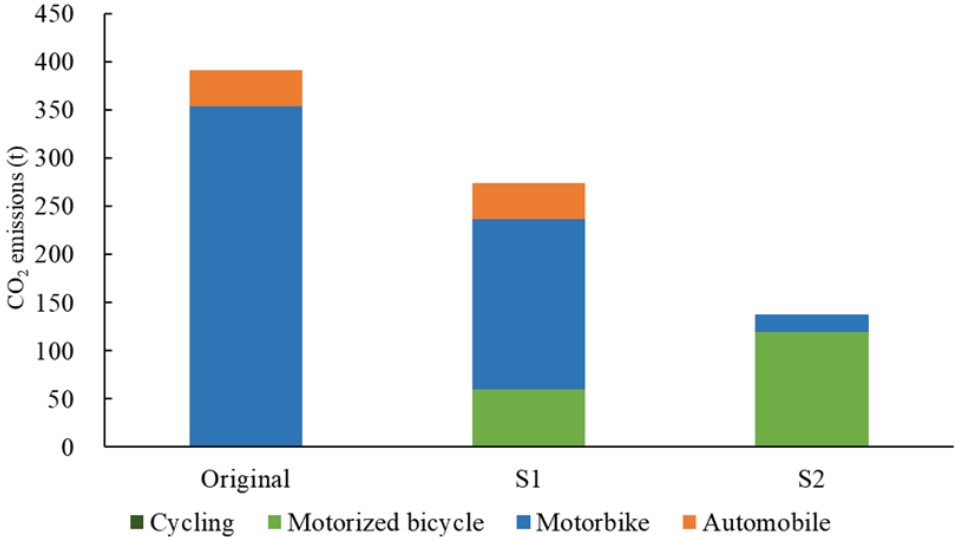

**Figure 7.** Differences in $CO_2$ emissions of OFD by changing the distribution.

Figure 7 indicates that $CO_2$ emissions from OFD can be reduced by shifting delivery method from high-emitting transportation to low-emitting transportation. When shifting from the original pattern to the S1 pattern, $CO_2$ emissions are reduced by 29.96%, while when shifting from the original pattern to the S2 pattern, $CO_2$ emissions drop by 64.89%, suggesting that a change in OFD distribution can contribute significantly to regional emissions reductions. Obviously, the best choice for OFD delivery is cycling for a short distance, followed by motorized bicycles for a relatively short distance. Thus, reconsidering the business area of the restaurants from a delivery perspective will contribute a lot to reducing $CO_2$ emissions. Moreover, a single OFD delivery to fulfill multiple orders will significantly alleviate the environmental impact of distribution.

### 5.3. Changing Plastic Containers

$CO_2$ emissions of PSP+PP, Bio-PE and PHBH using Japanese food were compared as an example. Due to data limitations, this analysis only considers the production of raw materials for plastic packaging containers and does not include the manufacturing process. Although Bio-PE is a bio-based plastic, it is not biodegradable, so incineration is considered

as the suitable waste disposal option [51]. On the other hand, PHBH is biodegradable, so it could be treated as plastic landfill waste when separated at the source.

Figure 8 shows that PHBH material has the highest $CO_2$ emissions, followed by Bio-PE. We can find that new plastic production has a much higher $CO_2$ emission than PSP+PP. In the plastic production stage, $CO_2$ emissions from PHBH production were 1.35 times those of PS. Although PHBH is a biodegradable material, most plastic disposal in Japan is via incineration [50]. The total $CO_2$ emissions of PHBH are higher than conventional materials (PSP+PP). Therefore, from the perspective of $CO_2$ emissions, PHBH and Bio-PE materials are not environmentally friendly in Japan.

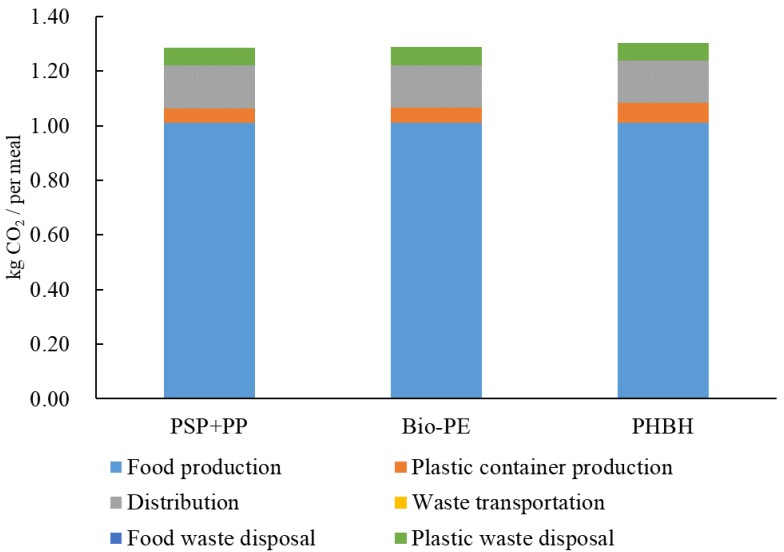

**Figure 8.** Differences in $CO_2$ emissions of OFD by changing the plastic containers.

*5.4. Limitations*

This study has designed a relatively comprehensive system framework using a meal as a functional unit, evaluating not only the $CO_2$ emissions associated with the production and handling of food products in OFD orders, but also the $CO_2$ emissions associated with the production and disposal of packaging materials used in OFD orders. However, as secondary data were used for both food production and packaging production, particularly as the food production data were not up to date, this may have led to an underestimation of the results. Additionally, items such spoons, chopsticks and napkins that may be associated with OFD orders were not considered.

Although $CO_2$ emissions are one of the most important indicators for evaluating environmental impact, a more comprehensive evaluation is needed to accurately assess the environmental impact of OFD, including considerations such as water pollution, $SO_x$ and $NO_x$. Future studies will consider more evaluation indicators, such as water pollution, $SO_x$ and $NO_x$.

**6. Conclusions**

This study evaluated the $CO_2$ emissions of an OFD order, considering multiple factors such as food options, types of plastic containers, delivery methods and distances, proportions of consumption, and methods of waste disposal. The scenario analysis could reveal how much each process contributes to the overall $CO_2$ emissions, providing directions for reducing $CO_2$ emissions from OFD. It showed that food production was the largest contributor to $CO_2$ emissions. Choosing environmentally friendly food options can significantly reduce the total $CO_2$ emissions of each OFD meal. On the other hand, when consumers leave leftovers, extra $CO_2$ emissions are generated during the production stage of the excess food, as well as during the collection, transportation and disposal of food waste. Therefore, if restaurants could provide more options for the volume of food and encourage customers

to order appropriate amounts of food, it would make a major contribution to reducing $CO_2$ emissions.

On the other hand, the total amount of plastic required for each OFD meal is relatively small, and the delivery distance is also relatively short in Japan, resulting in a smaller environmental impact from the food portion. However, initiatives such as improving delivery methods and using "green" containers can contribute to reducing $CO_2$ emissions from OFD.

There is still much research to be carried out. Our next step will involve conducting interviews with OFD platforms, restaurant owners and customers through surveys to gather real information and data. This will help us verify the results of our study and measure the actual amounts of food waste and plastic waste from OFD, allowing us to develop a more comprehensive environmental impact assessment of OFD.

**Author Contributions:** Conceptualization by C.L.; methodology, data collection and analysis and writing (original draft) by X.H., Q.Z. and C.L.; review and supervision by D.M. and C.L.; funding acquisition and project administration by C.L. All authors have read and agreed to the published version of the manuscript.

**Funding:** This research was funded by the Institute for Global Environmental Strategies (IGES) under the Strategic Research Fund 2021–2023 and supported by the Environment Research and Technology Development Fund (S-21) of the Environmental Restoration and Conservation Agency of Japan, 'Development of an Integrated Assessment Model linking Biodiversity and Socio-Economic Drivers, and its Social Application (IAM-B)' (2023–2027).

**Institutional Review Board Statement:** Not applicable.

**Informed Consent Statement:** Not applicable.

**Data Availability Statement:** Not applicable.

**Acknowledgments:** The authors would like to thank Emma Fushimi for proofreading this paper and would also like to express gratitude to the referees for their useful comments and suggestions.

**Conflicts of Interest:** The authors declare no conflict of interest.

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
