# Peer review of "Assessing CO2 Emissions of Online Food Delivery in Japan"

_sustainability, doi:10.3390/su15108084_

Round 1

Reviewer 1 Report

the paper is original and well structured, the results are well explained and clarified through graphs and tables. To increase the explanation of the method and methodology used

Reviewer 2 Report

Dear Authors!

You investigated a very interesting question which became even more important during the pandemic. To come to proper consclusions, many assumptions are necessary which are well documented in your paper. However, the problem of food residues in used boxes / packages is not adequately addressed. You discuss the problem of food residues in the context of marketing; I agree with your ideas. Therefore, I recommend to calculate a scenario with a fixed percentage of food residues as part of a sensitivity analysis.

Some more comments:

Line 92: However, paper boxes had the most serious environmental impact. Why? Due to their weight? But it's not from fossile origin.

Line 112: some food companies offer recyclable food containers, greatly enhancing the sustainability of food packaging. What does "recyclable" means in this case? Biodegradable? Compostable?

Table 6: 1. This table relates to transportation and not ro packaging material. This is the same heading as for Table 5. Similar problem with Table 7. Question: Fuel consumption is not the only GHG source of all vehicles. Carbon footprint also depends on the emissions linked to the materials. Please explain very shortly in the text why you only consider fuel.

Table 7: Which type of incineration did you enclose in your assessment? Normally, the plants produce energy; this decreases the net GHG emissions considerably.

Line 276 and beyond: This shows that the CO2 emissions from the production of packaging containers are negligible. - No, they are low as compared to food production, but not negligible. Furthermore, a scenario including food residues in the packaging is lacking (see above). In this scenario, landfilling can lead to serious GHG emissions (CH4).

Figure 6: Emissions from residual food waste should be taken into account (see above).

Line 372-377: These arguments sound strange. Please check the text again.

Line 413-414: Please see my annotation above: Food leftovers can play an important role and should be considered.

Reviewer 3 Report

please see file with comments

Round 2

Reviewer 2 Report

Thank you for your answers to my comments and suggestions.

I understand that this assessment and conclusions are strictly related to the conditions in Japan. In other countries (e.g. Western Europe), food residues would be either incinerated with energy recovery or digested to produce biogas. This would change the LCA completely.